# Bottleneck analysis of maternal and newborn health services in hard-to-reach areas of Bangladesh using 'TANAHASHI' framework': An explanatory mixed-method study

Mohiuddin Ahsanul Kabir Chowdhury[1]*, Farhana Karim[2], Mohammad Mehedi Hasan[2], Nazia Binte Ali[2], Abdullah Nurus Salam Khan[2], Md. Shahjahan Siraj[2], S. M. Monirul Ahasan[3], Dewan Md. Emdadul Hoque[4]

1 Asian University for Women, Chittagong, Bangladesh, 2 Maternal and Child Health Division, International Centre for Diarrhoeal Diseases Research, Bangladesh, Dhaka, Bangladesh, 3 World Vision, Cox's Bazar, Bangladesh, 4 United Nations Population Fund, Dhaka, Bangladesh

* ahsanul.chowdhury@auw.edu.bd

**Data Availability Statement:** Due to ethical restrictions related to protecting study participants privacy and confidentiality, data access is restricted

## Abstract

Maternal and Newborn Health (MNH) is of paramount importance in the realm of attaining sustainable development goals that also focuses on universal health coverage (UHC). The study aimed at identifying and exploring the bottlenecks in MNH services in Hard-to-reach (HtR) areas of Bangladesh using the Tanahashi framework exploring the possible remedial approaches. The study was conducted in four different types of HtR areas (hilly, coastal, lowlands, and river islands) by utilizing a sequential explanatory mixed-method design. Overall, we collected information from 20 health facilities and 2,989 households by interviewing 2,768 recently delivered women (RDW) with a structured questionnaire and qualitative interviews (n = 55) of facility managers, local stakeholders, RDWs, and health care providers (HCP). The quantitative data were analyzed principally for descriptive statistics and the qualitative data was analyzed by utilizing the thematic approach. Antenatal care, under-5 care, and family planning services were available in almost all the facilities. However, Normal vaginal deliveries were performed in 55.6% of the union-level facilities. Only 40% of sub-district level facilities had provision for C-sections. Blood transfusion services were available in only 20.1% of facilities, whereas laboratory services were obtainable in 51.7% of facilities. Overall, the bottlenecks were identified in cases of availability of drugs, human resources, transportation, lack of knowledge regarding different essential services and health components, out of pocket expenditure etc. There have been several remedial approaches suggested from both the demand and supply side that included incentives for care providers for staying in these areas, a coordinated transport/referral system, and health education campaigns. More research works are warranted in HtR areas, especially to test the proposed interventions. Meanwhile, the government should take the necessary steps to overcome the bottlenecks identified.

by the Ethical Review Committee of icddr,b. According to the icddr,b data policy (http://www. icddrb.org/policies), interested parties may contact Ms. Armana Ahmed (aahmed@icddrb.org) with further inquiries related to data access.

**Funding:** The research study was funded by the Swedish International Development Cooperation Agency (Sida) through International Centre for Diarrhoeal Diseases Research, Bangladesh (Grant number: GR 01455). MAKC received funding for this study. URL of funder website: https://www. sida.se/English/. The funders had no role in study design, data collection and analysis, decision to publish, or preparation of the manuscript.

**Competing interests:** The authors have declared that no competing interests exist.

# Introduction

Despite the astonishing achievement of decreasing the maternal mortality ratio (MMR) from 342 to 211 per 100,000 live births between 2000 and 2017, still annually over 800 women worldwide are dying from pregnancy and childbirth-related complications [1]. Maternal deaths during pregnancy and childbirth are more prominent in low and middle-income countries [2]. Similarly, during the same time, although the global neonatal mortality rate (NMR) was reduced by 51% the estimated number of annual neonatal deaths is about 2.5 million [3]. Almost 70% of these deaths occur in low-income countries of Africa and Asia [4]. In this context, to foster the progress worldwide, World Health Organization in the General Assembly, adopted the resolution of sustainable development goals which include the agenda to reduce both maternal mortality ratio and neonatal mortality rate [5].

In Bangladesh, there has been considerable progress in the reduction of maternal and neonatal mortality in recent decades [2, 6]. For instance, the MMR was reduced by almost 70% between 1990–2017 [7], from 574 to 173 per 100000 live births with an average annual declining rate of 4.35%. If Bangladesh sustains this rate, by the year 2030, the MMR of the country would be 97 per 100,000 livebirths while to reach the global target for MMR of 70 per 100,000 livebirths, the rate needs to be accelerated to 8.3% [1, 8]. On the other hand, Bangladesh has successfully reduced the NMR from 52 to 17.5 deaths per 1000 livebirths from 1994–2020 with an annual declining rate of 4.1%, which if maintained, will enable the country to reach the targeted SDG of less than 12 per 1000 livebirths [9–11]. However, many recent studies suggested that the maternal and neonatal mortality rates have increased in many developing countries during the Covid-19 pandemic [12, 13]. This will put up a challenge for the countries to achieve the SDGs in time regarding MMR and NMR along with warranting universal health coverage (UHC).

To ensure full coverage through UHC packages for maternal and newborn health (MNH) a strong, efficient, well-functioning health system is required to ensure availability, easy accessibility, acceptability, and effective utilization of existing health services by the poor population without facing any financial hardship [14]. In the last few decades, the government has focused on the improvement of Primary Health Care (PHC) infrastructure has sought the attention of the government [15, 16], but yet, challenges exist regarding health of the people living in the Hard to Reach (HtR) areas like insufficient health professionals and medical logistics, which causes high maternal and child mortality, malnutrition, and other infections, as suggested by the studies conducted in those regions, and gaps exist in comparison to the recent most national representative survey for maternal and newborn health indicators [9, 14, 17–19]. For instance, the immunization rate among the children aged 12–23 months was 44% in comparison to the national coverage of 71%, as documented by a study conducted in 2009 [20]. A recent district-level survey, conducted to assess socio-demographic and health care utilization, showed poor utilization of the maternal and newborn care interventions in the HtR districts like Bandarban, Sunamganj, Kurigram, Satkhira etc, in comparison to the national average [21]. Since the Government of Bangladesh is emphasizing both the MNH sector and UHC, it is time to blend the paradigms and look to achieve UHC in the MNH sector first and set up an ideal example for other components of health care.

Although the shortage of modern medical facilities, continuous absence of qualified health care staff, and lack of basic medicines and medical equipment, might be the common issues to establish quality MNH care in HtR areas, limited data are available for these areas that are required to design and implement appropriate policy [14]. Moreover, the in-depth explorations are lacking for the reasons behind the health system failure and the possible solutions for those obstacles. This study aimed at identifying the bottlenecks in MNH services in HtR areas

of Bangladesh using the Tanahashi framework and exploring the possible remedial approaches for the identified bottlenecks.

## Materials and methods

### Study design

We employed an explanatory sequential mixed method design in this study [22]. At first, we conducted a quantitative assessment to identify which domain of the Tanahashi framework [Fig 1] possesses the bottlenecks and later, in the subsequent phase, we explored what those bottlenecks along with possible remedies through a qualitative approach. The Tanahashi framework is explained in S1 File.

### Study setting

The study involved 4 types of HtR areas of Bangladesh namely coastal (Southern region), Hilly (South-Eastern region), Lowlands/*haors* (North-Eastern region), and River islands/*chars* (Northern region). We conducted a household survey in 36 villages of 12 unions from 4 districts (1 of each type of HtR area) out of 98 villages in total. We also conducted a health facility survey to assess the availability domain of the Tanahashi framework in all the functioning facilities (N = 29) in our study regions of which 10 were Upazila Health Complexes—UHC (Subdistrict level health facility), 9 Union Health and Family Welfare Centres—UHFWC(Union level facilities), and 10 community clinics—CC (village level facilities). Each CC is meant to serve three villages, and in our study area, we found 10 functioning CCs.

### Participants and sample size

Our study participants included 2,768 recently delivered women (RDW) of 2,989 households from whom we collected information related to accessibility, utilization, adequate coverage, and effective coverage. Recently delivered women here were women who delivered their babies within the previous year of the data collection period. In addition, to extract information from all other domains, we interviewed 6 facility managers, 10 local stakeholders, 17 Health care providers (HCP), and 22 RDWs in-depth. These numbers seem to be sufficient for qualitative themes since Guest et al. (2008) showed that 7–12 qualitative interviews will reach the data saturation in more than 95% of occasions [23], and data saturation refers to the point in the qualitative research when no new information or new theme is being emerged during in data collection [24]. Table 1 shows our sample size for different components.

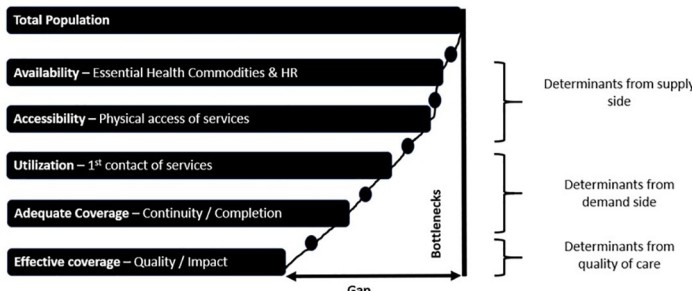

**Fig 1. Tanahashi model to identify the health system bottlenecks.**

**Table 1. Data collection methods and sample size of the study participants.**

| Data collection methods | Type of study participants | Type of data collection tools | Number of participants |
|---|---|---|---|
| Health facility observation | N/A | Quantitative | 29 |
| Structured interview | Recently delivered women | Quantitative | 2,768 |
| Key informant interview | Health managers | Qualitative | 6 |
| Key informant interview | Stake holders | Qualitative | 10 |
| In-depth interview | Health care providers | Qualitative | 17 |
| In-depth interview | Recently delivered women | Qualitative | 22 |
| Focus group discussion | Community people | Qualitative | 12 |

## Sampling and data collection technique

In this study, for quantitative data collection, we used multi-stage simple random sampling technique to select the villages. At first, we generated lists of districts having HtR areas and then randomly selected one district for each type of HtR areas. Afterward, we generated lists of HtR unions of each of those districts and randomly selected three (3) unions from each district and ended up with 12 unions. A list of the unions of the selected districts is provided in S2 File. Finally, we randomly selected three (3) villages from each union and the community component of our study took place in those 36 villages. At this stage, we utilized a systematic sampling method to select the households. Our data collectors reached the center of the village and then randomly chose a direction by the pen-spinning method. Then, the data collectors visited every alternate household to collect information regarding RDWs. The data collectors were advised to turn right once they reached the dead-end of the village. For the qualitative component, we used the snowball sampling method to have our study population. At first, we visited the local stakeholders and from them, we got the names and addresses of potential study participants. The snowball sampling was used to get save resources by knowing about the study participants earlier and going to them directly for collecting data. We selected the other qualitative participants (facility managers, health care providers, community people) through purposive sampling.

## Data collection tools

We utilized a direct observation checklist for facility assessment and a household listing form with the structured questionnaire for the community survey. For the facility survey, we used the adapted version of "Bangladesh Health Facility Survey 2014" questionnaire [25], and for the community survey we prepared a structured questionnaire adapting the "Bangladesh Demographic Health Survey 2014" household questionnaire [26]. We also had key informant interview (KII) guidelines for facility managers and community stakeholders in order to learn about the bottlenecks in-depth. Besides, we had in-depth interview (IDI) guidelines for HCPs and RDWs. In addition, we employed focus group discussion (FGD) guideline for the community. These different qualitative tools are required to achieve our research objectives. For instance, the KIIs of facility managers provided us with information regarding the availability domain of Tanahashi framework and service utilization from the supply side perspective. From the KII of the local stakeholders, we extracted information regarding the bottlenecks related to the accessibility domain. The IDIs of the RDWs gave us their lived experiences for utilization and coverage of the MNH care. Finally, the FGDs helped us to identify the bottlenecks and possible remedial strategies from demand-side perspectives, Moreover, these different approaches of data collection also helped us in data triangulation. The data collection tools

(Both English and Bangla, except for the facility survey, which was not translated) are provided as S3 File.

### Quantitative variables

Among the quantitative variables, we had variables related to the health facilities and variables related to the RDWs. The facility-related variables were availability of services, availability of equipment required to provide maternal and newborn health services, the proportion of posts for HCPs filled, and training of the HCPs. On the other hand, the variables related RDWs included variables related to Antenatal care (ANC), care during delivery, and postnatal care (PNC). The ANC-related variables were the number of ANC visits, place of ANC, service provider of ANC, nutritional supplement provided during ANC, counseling during ANC, etc. The variables for care during delivery contained place, type, service provider, complications, expenditure, and so forth. Lastly, number of PNC visits, service providers, counseling, etc. were included as PNC-related variables. For the variables, we developed an indicator matrix where the domains of Tanahashi framework for different variables were defined [S4 File].

### Qualitative themes

Two major qualitative themes were considered. The first major theme was bottlenecks of MNH health services which were sub-thematically divided according to different domains of the Tanahashi framework. The second major theme was the remedial approaches for the improvement of the services and the sub-themes were approached from demand side, supply side, and the policymaker's perspective.

### Data collectors' training and data collection

Three teams of data collectors were formed of which two were for the quantitative component for conducting the facility survey and community survey respectively. The other team was responsible for the qualitative component. The facility survey comprised of four medical graduates with prior experience in conducting the facility survey. This team was led by FK, one of the study co-investigators. The community survey team was composed of 20 Field Research Assistants (FRA) of whom 10 were for household listing, and the other 10 were for community survey. The FRAs were university graduates from different disciplines with previous experience in data collection at community-level. This team was supervised by a Field Research Officer (FRO) who maintained communication with the central team through MMH, another study co-investigator. Finally, the qualitative team was led by SMMA, the qualitative lead of our study, who had 4 FRAs from Anthropology backgrounds under him. All the data collectors' were intensively trained on the data collection tools and guides followed by pre-testing in a district nearby Dhaka, the capital of Bangladesh. Then another daylong session was conducted to discuss the issues raised during pre-testing, after which we finalized our data collection instruments. The data collection was conducted in four HtR zones one after another, and in each zone, data collection started with the quantitative component first. The household listing team went to the community while the facility team started their also started their survey. Later on, after finalization of household listing, the study population for the zone is finalized for the community survey, and the other half of the community survey team conducted the household survey. Meanwhile, the qualitative team conducted the interviews in the facility first, and afterwards headed to the community to finish the rest. The similar approach was repeated in all the HtR zones for data collection.

## Data analysis methods

Range and consistency checks for the quantitative data were conducted during data entry process into SQL server. The cleaned data were then transferred into Stata 13. For analysis, the descriptive statistics were utilized principally. We utilized frequency distribution for the categorical variables and measures of central tendency like mean and standard deviation for the quantitative variables. Data is represented by appropriate numerical, tabular and graphical methods. The data have been analyzed utilizing the Tanahashi framework i.e. at the levels of availability, accessibility, utilization, adequate coverage, and effective coverage which are presented as proportions of the total population as per the definition provided in the indicator matrix. All Key Informants Interviews (KII) and Focus Group Discussions (FGD) were digitally recorded after taking the consent from the participants and then transcribed. The availability for the Tanahashi framework was calculated through a composite indicator built from the scoring of availability of services, availability of human resources, their training status, and stock-outs of relevant drugs and equipment.

Besides, all the qualitative interviews and interview-notes for 16 KIIs, 39 IDIs, and 1 FGD were placed in organized transcripts which were randomly checked against audio recording to ensure quality. Researchers (3 members of the Qualitative team) read all transcripts to achieve data familiarization. Transcripts were then be reviewed by investigators for obvious errors, and missing texts, and revised if necessary. Then the significant statements and phrases were extracted from each transcript. A primary codebook was developed in consultation with the investigators [S5 File]. Manual coding was done by the investigators. At the beginning, a minimal set of interviews was coded by 2 different researchers to check for inter-coder reliability. At this stage, a third researcher checked the codes from these two and looked for differences. If there were any differences, those were discussed and clarified. Upon establishing the inter-coder reliability, a master code list was generated, based on which the rest of the transcripts were coded. Then the transcripts were organized into themes which evolved into theme clusters and eventually into theme categories. Thus, the qualitative data analysis was performed through thematic approach.

## Ethical consideration

The study was approved by the Institutional Review Board (IRB) of icddr,b (Grant no. 01455). As per the protocol, informed written consents in the local language were taken from the study participants before data collection. Regarding participants who were illiterate, their thumb impression was collected as consent before conducting an interview. However, for the FGDs, we had verbal consent from the participants, and it was recorded.

## Results

### Sociodemographic characteristics

The mean age of 2,768 recently delivered women was 25.4 years (±5.8 years). Although only 6.5% were working women, about 96% had their personal mobile phone. More than half of the participants had to go the nearest UHC on foot. Table 2 summarizes the sociodemographic characteristics of the women.

### Availability of service provision in the facilities

ANC was provided in all of the 29 facilities we surveyed. However, Normal vaginal delivery (NVD) was performed in about half of the facilities. Moreover, Caesarean section (C/S) was

Table 2. Sociodemographic characteristics of the study participants [N = 2,768].

| Criteria | Mean / n | SD / percentage |
|---|---|---|
| Age (in years) | 25.4 | 5.8 |
| Ever been to school | 2355 | 85.1% |
| **Level of education:** | | |
| Primary | 947 | 40.2% |
| Secondary | 1249 | 53.0% |
| Higher Secondary or above | 159 | 6.8% |
| **Religion:** | | |
| Muslim | 2245 | 81.1% |
| Hindu | 424 | 15.3% |
| Others | 99 | 3.6% |
| **Occupation:** | | |
| Housewife | 2587 | 93.5% |
| Agriculture | 181 | 6.5% |
| **Owning a mobile phone:** | | |
| Yes | 2660 | 96.1% |
| No | 108 | 3.9% |
| **Mode of transport to nearest UHC:** | | |
| On foot | 1503 | 54.3% |
| By any vehicle | 395 | 14.3% |
| Both | 870 | 31.4% |

practiced only in 40% of the designated facilities and the blood transfusion provision are available in 60% facilities. Table 3 provides the summary of service provision in the health facilities.

## Availability of human resources in study health facilities

We collected data regarding number of posts for different types of health providers in each health facility. In UHCs, only 30% obstetrics and gynecologist consultants were posted, while no pediatric consultant was posted during data collection period. Against 71 sanctioned posts of medical officers and 175 posts of nurses, only 61% posts and 68% were filled, respectively. Only in two UHCs, anesthetists were posted. Only 40% midwives were posted against 38 sanctioned posts in 10 UHCs. In case of UHFWC, Family Welfare Visitors (FWV) and Sub Assistant Community Medical Officers (SACMO) were posted in seven out of nine UHFWC

Table 3. MNH care provision in the public health facilities of HtR areas of Bangladesh in percentage.

| MNH care components | UHC (n = 10) | UHFWC (n = 9) | CC (n = 10) | Total |
|---|---|---|---|---|
| Antenatal care | 100% | 100% | 100% | 100% |
| Laboratory services | 80% | 11.1% | 60% | 51.7% |
| In-patient care | 100% | N/A | N/A | 100% |
| Normal vaginal delivery | 100% | 55.6% | N/A 0% | 78.9% |
| Caesarean section | 40% | N/A | N/A | 40.0% |
| Blood transfusion services | 60% | N/A | N/A | 60.0% |
| Emergency management | 100% | N/A | N/A | 100% |
| Postnatal care | 100% | 100% | 100% | 100% |
| Family planning services | 100% | 100% | 70% | 89.7% |
| Under-5 care | 100% | 100% | 100% | 100% |

Table 4. Available human resources against sanctioned post in the study facilities.

| Upazila Health complex (N = 10) | Sanctioned post | % of filled post |
|---|---|---|
| **Upazila Health complex (N = 10)** | | |
| Consultant (OBS. & GYN.) | 10 | 30.0 |
| Consultant (Pediatrics) | 7 | 0.0 |
| Medical Officer (Total) | 71 | 60.6 |
| Staff nurses (Total) | 175 | 68.0 |
| Anesthetist | 10 | 20.0 |
| Midwives | 38 | 39.5 |
| Medical assistants | 20 | 90.0 |
| Family Welfare Visitor (FWV) | 13 | 84.6 |
| **UHFWC (N = 9)** | | |
| Medical assistants/ SACMO (Total) | 9 | 77.8 |
| Family Welfare Visitor (FWV) | 9 | 77.8 |
| **Community clinic (N = 10)** | | |
| Community Health Care Provider (CHCP) | 10 | 100.0 |
| Family Welfare Assistant (FWA) | 10 | 60.0 |

during data collection period. On the other hand, each CC had Community Health Care Provider (CHCP)during data collection post. The results are summarized in Table 4.

## MNH care utilization information

At least one ANC was availed by 75.4% although only 28.6%had four or more ANC visits. About two-thirds of the respondents had Tetanus prophylaxis, Iron, Folic acid, and Calcium supplements during ANC. About 69% of the RDWs delivered their baby at home. Most (85%) of the women experienced normal vaginal delivery. Almost half of the respondents had complications of whom 948 (71.3%) were treated. However, the expenditure was mostly (99.2%) carried out by the patients themselves. The immediate newborn care guidelines were not followed in many cases as depicted in Table 5.

## Tanahashi framework analysis

We conducted Tanahashi framework analysis for Antenatal care, Iron and Folic acid (IFA) supplements during ANC, Delivery care, and Postnatal care as shown in Fig 2(A)–2(D). The definition for each domain is provided in the indicator matrix [S4 File]. For ANC, the availability was calculated as 75%. The accessibility, as defined as the "Proportion of women aged 15–49 who had a live birth in past one year, lives within 1-hour travel to UH&FWC/ union sub-center", was 50.3%. So, regarding the determinants from the supply side there seem to be several bottlenecks since only half of the population could reach the facility. However, the existence of bottlenecks in other domains was also identified since utilization, adequate coverage, and effective coverage were 33.5%, 13.6%, and 0% respectively (Fig 2A). Similarly, bottlenecks were present in availability (79.3%), accessibility (50.3%), utilization (33%) and adequate coverage (8.9%) domains for IFA supplementations during ANC although it seems that there was almost no bottleneck for effective coverage (Fig 2B). Regarding NVD, the biggest bottlenecks should be in availability and utilization domain as the graph drops abruptly (from 100% to 60% in availability and from 50.3% to 16.9% in case of utilization] whereas mild to moderate bottlenecks were identified for accessibility and effective coverage. However, adequate

**Table 5. Information related to MNH care among the study participants.**

| Criteria | n | percentage |
|---|---|---|
| **Antenatal care:** | | |
| None | 681 | 24.6% |
| At least one | 2087 | 75.4% |
| Four or more | 792 | 28.6% |
| **Prophylaxis and supplements during ANC:** | | |
| Had Tetanus Toxoid (TT) during pregnancy | 1711 | 61.8% |
| Took Iron and Folic Acid (IFA) tablets | 1970 | 71.2% |
| Took Calcium supplements | 1877 | 67.8% |
| **Place of delivery:** | | |
| Home | 1918 | 69.3% |
| Public health facility | 343 | 12.4% |
| Private health facility | 479 | 17.3% |
| Others | 28 | 1,0% |
| **Mode of delivery:** | | |
| Normal vaginal delivery | 2353 | 85.0% |
| Caesarean section | 415 | 15.0% |
| **Complications during delivery:** | | |
| Yes | 1329 | 48.0% |
| No | 1439 | 52.0% |
| **Treated for complications [n = 1329]:** | | |
| Yes | 948 | 71.3% |
| No | 381 | 28.7% |
| **Expenditure for treating complications [n = 948]:** | | |
| Government | 8 | 0.8% |
| Out of pocket | 940 | 99.2% |
| *__Drying and wrapping before delivery of placenta [n = 2739]:__* | | |
| Dried | 404 | 14.7% |
| Wrapped | 419 | 15.3% |
| *__Appropriate cord care [n = 2739]:__* | | |
| Cord cut by sharp sterile instrument | 2543 | 92.8% |
| Nothing applied to cord | 1692 | 61.8% |
| *__Feeding related indicators [n = 2739]:__* | | |
| Breastfed before delivery of placenta | 63 | 2.3% |
| Colostrum was provided | 2433 | 89.4% |
| Only breastfeeding in the 1st 3 days | 2018 | 73.7% |

*Information collected from the mothers giving live births only.

coverage had least bottleneck as depicted in Fig 2C. Finally, for PNC, bottlenecks were evident at all the domains except for adequate coverage as shown in Fig 2D.

## Identified bottlenecks from qualitative interviews

**Supply side bottlenecks.** *Availability*. The mostly stated bottleneck was the unavailability of a sufficient health workforce in the health facilities. The facility managers and HCPs accepted the fact that there is scarcity of doctors:

**Fig 2. (A-D) Tanahashi framework analysis for ANC, Delivery care, and PNC.**

*"Doctors do not want to join or stay in these remote facilities"*

*–[KII: Facility manager 3]*

This was also echoed by the community people and leaders:

*"The people do not find doctors, most of the time they are treated by sisters (Nurses)"*

*–[KII: Community Stakeholder 1]*

The facility managers also reported stock out of drugs, unavailability of training facilities, lack of training, and unavailability of laboratory and blood transfusion services as important bottlenecks.

*"Some medicines get stocked out often, while some others pass their expiry date unused"*

*–[KII: Facility manager 2]*

However, the people were concerned about the unavailability of emergency services like C-section and laboratory services in the facilities.

*"All the services, such as C-section are not available in the nearby facility. We have to go to the district hospital for that."*

*–a focus group participant*

The RDWs also talked about being ignorant about several health services and they also felt that proper counselling is missed in those facilities. Some RDWs also mentioned about unavailability of medicine in the facility that bounds them to by those from outside.

*"We do not get all the drugs from the facility, and we also need to do the investigations from outside"*

–[IDI: RDW 13]

*Accessibility.* Transportation and communication have been mentioned by most of the study participants. The facility managers and HCPs stated that transportation is the main issue behind poor accessibility. The community people also mentioned the cost of transportation as another reason since generally, they are poor.

*"We are poor, we cannot spend so much on health"*

–[IDI: RDW 8]

Some of the RDWs mentioned that it is very tough to the facilities on foot when they have any emergency. Therefore, they prefer home delivery to facility delivery.

**Demand side bottlenecks.** *Utilization and coverage.* The facility manager mentioned the lack of awareness of the community people and training gaps among the HCPs as the main bottleneck for poor utilization and coverage. One of the facility managers also cited the lack of motivation of the HCPs to work and stay in the remote area as a potential bottleneck for adequate coverage. The RDW, again, mentioned out-of-pocket expenditure as a bottleneck for utilization of services. They also added their knowledge gap about health services.

*"We are ignorant about several health services, or what is good or bad for health. If the apas (CHWs) tell us, we come to know"*

–[IDI: RDW 11]

Table 6 summarizes the identified bottlenecks for different domains of Tanahashi framework.

*Proposed remedial approaches.* Several remedial approaches were proposed for different bottlenecks. The HCPs who work and stay in HtR areas could be incentivized financially, as suggested by a couple of community stakeholders. The facility managers also stated about incentives in order to warrant the stay of the HCPs in HtR areas.

**Table 6. Identified bottlenecks for different domains of Tanahashi framework.**

| Tanahashi Domains | Identified bottlenecks |
|---|---|
| Availability | • Inadequate human resources as health workforce<br>• Frequent stock out of drugs and other resources<br>• Inadequate training of HCPs<br>• Unavailability of laboratory facilities |
| Accessibility | • Poor transportation and communication system<br>• High expenditure for availing transportation |
| Utilization | • Lack of awareness and Ignorance about the available services<br>• Cultural inclination towards home delivery |
| Adequate and Effective Coverage | • Gaps in training and lack of motivation of HCPs<br>• Out of pocket expenditure |

*"Government can arrange incentives for the doctors for serving in these HtR areas"*

*–[KII: Community stakeholder 3]*

To ensure better availability the facility managers proposed to plan and list the drugs systematically according to their utilization rate so that stock out can be prevented. They, along with the HCPs, also advocated for arranging regular and refresher training for the providers on regular basis.

*"We need to list the drugs according to their utilization rate and send the demand slip for longer period"*

*–[KII: Facility Manager 4]*

Besides ensuring the availability of the HCPs in the facilities, the RDWs also emphasized about laboratory facilities and health education campaigns which would help to achieve utilization and coverage domains.

*"We do not know the important things; we are eager to know what is important for our health and our kids' health."*

*–[IDI: RDW 2]*

During the focus group discussion, an interesting solution of the accessibility issues was proposed by one of the participants and was later endorsed by other participants. They discussed the possibility of the development of an integrated community referral/transport system where the administrative and health wings will be coordinated with the community to ensure safe, early transfer of the patients. In this system, designated transportations will be kept ready at the community for the seriously ill patients; the cost of which will be borne by the local government or the community stakeholders. The health facilities will be informed of the patients so that they get prepared beforehand. Thus, a lot of bottlenecks could be solved with a single intervention.

*"It would be great if there is a centrally coordinated (by local government and health facility) community referral system so that there is no delay in the treatment"*

*–a focus group participant*

## Discussion

In this study, more than 85% of the RDWs ever attended school and only about 7% had more than 10 years of education. Among all the RDWs, 93.5% were housewives and 96.1% owned mobile phones. No other positions than CHCP in the health facilities were filled according to the sanctioned posts. Less than one-third of the participants had 4 or more ANC visits during their pregnancy and only 12.4% were delivered in the public health facility. About 70% of the RDWs opted for home delivery for their childbirth. About half of the study participants reported of having complications during pregnancy and childbirth, however, 28.7% of them did not go for treatment. Quality of care for immediate newborn care documented mixed results where about 15% of the babies were dried and wrapped before delivery and placenta, in 61.8% of the babies nothing was applied to the cord, and colostrum was fed to 89.4% of the babies. However, only breastmilk was provided to 73.7% of babies in the 1st 3 days of their life.

Tanahashi framework analysis showed that the major bottlenecks were for availability, accessibility, and utilization domains. The key bottlenecks identified from the qualitative component were inadequate human and other resources, poor communication, and transportation, out-of-pocket expenditure, lack of awareness of services among the community, lack of motivation of the HCPs to stay at HtR areas, and training gaps. Some remedial approaches were also suggested both from the demand and supply sides.

Around 85% of the RDWs in this study attended schools which is much higher than the literacy rate of Bangladesh of 70% [27]. However, this goes along with the increasing trend of the girls' education level in Bangladesh which is being continuously encouraged by the government for the last couple of decades [27]. Another striking feature of this study was owning mobile phones by the study participants which was over 96%. However, this is also consistent with the national mobile usage subscription which was about 124.7 million in 2015 [28]. This high usage of mobile phones paved the path for many m-health-based interventions in the country [28]. We documented the unavailability of adequate human resources in the health facilities. Human resources management has always been a challenge for Bangladesh since the doctors and nurses prefer to stay closer to the bigger cities due to the better amenities and infrastructure in comparison to HtR areas [29].

The community clinics (CC) and UHFWCs are not entitled for performing C-sections and blood transfusion, and the CCs do not provide NVD services as well. However, more than half of the UHFWCs reported of conducting NVDs which is a good in comparison to the national standard of Bangladesh. Yet, the C-section (40%) and Blood transfusion services (60%) in UHCs are alarming since those are the places where the community people go if any complication arises. Moreover, these HtR areas have poor communication systems which cause them to travel more with complicated patients and thus, results in undue delays. It is also worth mentioning that even many designated emergency obstetric care (EmOC) facilities are not functioning in Bangladesh because of human resources and logistics issues, let alone these HtR facilities [30]. A recent research conducted in Kurigram, a district of this study, found similar results for the service provision in the facilities [31].

Bangladesh Demography Health Survey (BDHS) 2017–18 found the percentage of women receiving at least one ANC to be 82% which is about 6 percentage points higher than our study [9]. This subtle difference is expected since our study was conducted in HtR areas. Contrariwise, nationally, 47% of women had four or more ANCs which was much higher than what we got in our study (28.6%) [9]. In comparison to BDHS 2017–18, less mothers had facility delivery (30% vs. 50%) which can again be explained by the lack of a good communication system in HtR areas [9]. Again, a similar pattern was seen in postnatal care when it expectedly corresponded to facility delivery percentage [9]. Lastly, the proportions of practicing essential newborn care were low in both nationally and in our study.

Our study is one of the few studies that utilized the Tanahashi framework in the context of Bangladesh. We found low effective coverage for the indicators we used for analyzing MNH care. The proportions for availability and accessibility together were around 50% which was also not equal to the standards. A multi-country study conducted in Bangladesh, Haiti, Malawi, Nepal, Senegal and Tanzania also ended up with similar results like ours when they used the same framework [32]. These findings flag the importance of looking into the areas of bottlenecks by the relevant public and private stakeholders. Interestingly, the bottlenecks identified through qualitative exploration were not uncommon. In different studies, the same bottlenecks were found which have been aggravated in our study on account of the remoteness of the study sites [9, 31–34].

This current study identified inadequate human resources, frequent stock out of drugs, inadequate training of HCPs, and unavailability of functioning laboratory facilities as the

major bottlenecks for the availability domain of the Tanahashi framework. Many studies conducted in many countries reported the same [35–38]. For instance, a multi-country analysis conducted in 12 countries of Asia and Africa identified the health workforce as the most critical health system bottleneck [35]. Many other studies reported lack of trained health care providers as a health system challenges for maternal and newborn health like our study [35, 39–41]. A few of these studies also identified frequent stockout and lack of functioning laboratory facilities as major bottlenecks [35, 41].

Poor communication system has always been a challenge for accessing health services in the hard-to-reach areas as evidenced by the studies conducted in low-and-middle-income countries of Africa and Asia [42–44]. However, this challenge can be somewhat mitigated by utilizing the m-health technologies to provide reminders and health education to pregnant women and their caregivers. A study has shown the feasibility of using mobile phones for improving vaccination coverage in Bangladesh which can be replicated for other situations also [45]. The other major bottleneck for the accessibility domain was high expenditure for transportation to and from the nearest health facility, especially during any emergency situation. Studies conducted in some African countries like Ghana, Malawi, Sierra Leone, and Nigeria mimed this finding [46–49].

Lack of awareness and ignorance regarding available health services were identified as the bottlenecks for utilization of MNH care which goes along with the findings of the studies that looked for the barriers to health services utilization in many countries. For instance, a study investigated the correlates and barriers to the utilization of health services for delivery in South Asia and Sub-Saharan Africa identified lack of awareness and knowledge as major barriers to health service utilization [50]. In the countries like Bangladesh, the social stigma and cultural norms also act as barriers to health care utilization [51–54], which has also been reported in our study.

We identified the training gaps of the HCPs as a major bottleneck for adequate and effective coverage. A study in Ethiopia also resonated with this finding while evaluating barriers to providing quality emergency obstetric care [55]. Similar findings have also been reported in Kenya [56] and Nepal [57]. Out of pocket expenditure for health is catastrophic in many country settings [58–60], and our qualitative exploration has identified this as a major hurdle against adequate and effective coverage of MNH care utilization.

We have also documented the remedial approaches for the identified bottlenecks from the demand side perspective which will open a window for the relevant stakeholders and policymakers to see from the users' view and design the forthcoming interventions accordingly.

## Strengths and limitations

The uniqueness of our study is laid on its qualitative exploration of remedial approaches from both demand and supply-side perspectives. The interventions suggested by the community have the potentials to succeed in the long run. The idea of an 'integrated community referral/ transport system' could be a milestone to solve the health-related problems in HtR areas. Moreover, this could be implemented in the other developing countries in need as well. Rigorous quality assurance in tools development, training, and data collection were the other notable strength of this study. Finally, the range of data collection methods employed in this study has enabled the researchers to triangulate the findings and strengthen their study.

The descriptive statistical analysis pattern of the Tanahashi framework is a limitation of this study. Nevertheless, the data has been triangulated by a qualitative approach to decipher this problem. Another limitation was the issue of generalizability since the study was restricted to the HtR areas only. However, because a lot of studies have been conducted on the overall

health system of Bangladesh, the researchers of this study were not concerned about the generalizability issue; rather they emphasized the findings of HtR areas only. Because of the budget constraints, the information could not be taken from all parts of the HtR areas, rather we had to select one district from each type of area, which can be listed as one of our limitations as well. Another limitation was the possible recall bias when the information was collected from the RDWs regarding their experience during pregnancy and childbirth.

## Conclusion

Since limited evidence have been generated to date regarding the health system bottlenecks in HtR areas, more research works are warranted. Besides, to achieve SDGs, the government should take the findings of this study into account and take necessary steps to improve the MNH care in HtR areas along with the other parts of the country. The findings may help the country to accelerate its progress in reducing maternal and neonatal mortality and can achieve the relevant SDGs 3.1 and 3.2 in time.

## Supporting information

**S1 File. Tanahashi framework_LTC.**
(DOCX)

**S2 File. List of unions_LTC.**
(DOCX)

**S3 File. Data collection tools.**
(PDF)

**S4 File. Indicator matrix_LTC.**
(DOCX)

**S5 File. Codebook_LTC.**
(DOC)

## Acknowledgments

The authors would like to acknowledge the support from the Swedish International Development Cooperation Agency (Sida). This manuscript was developed when the lead author MAKC was in BRAC James P Grant School of Public Health and SMMA was in International Rescue Committee. The authors also acknowledge the efforts from the data collectors and the field managers who travelled to the HtR areas and collected data to accomplish this study.

## Author Contributions

**Conceptualization:** Mohiuddin Ahsanul Kabir Chowdhury, Nazia Binte Ali, Dewan Md. Emdadul Hoque.

**Formal analysis:** Mohiuddin Ahsanul Kabir Chowdhury, Mohammad Mehedi Hasan, Md. Shahjahan Siraj, S. M. Monirul Ahasan.

**Funding acquisition:** Mohiuddin Ahsanul Kabir Chowdhury.

**Investigation:** Mohiuddin Ahsanul Kabir Chowdhury, Farhana Karim, Nazia Binte Ali, Abdullah Nurus Salam Khan, S. M. Monirul Ahasan.

**Methodology:** Mohiuddin Ahsanul Kabir Chowdhury, Nazia Binte Ali, Abdullah Nurus Salam Khan, Dewan Md. Emdadul Hoque.

**Project administration:** Mohiuddin Ahsanul Kabir Chowdhury, Farhana Karim, Mohammad Mehedi Hasan, Abdullah Nurus Salam Khan, Md. Shahjahan Siraj, S. M. Monirul Ahasan.

**Resources:** Mohiuddin Ahsanul Kabir Chowdhury, Farhana Karim.

**Software:** Farhana Karim.

**Supervision:** Mohiuddin Ahsanul Kabir Chowdhury, Farhana Karim, Mohammad Mehedi Hasan, Nazia Binte Ali, Md. Shahjahan Siraj, Dewan Md. Emdadul Hoque.

**Visualization:** Mohiuddin Ahsanul Kabir Chowdhury, Farhana Karim.

**Writing – original draft:** Mohiuddin Ahsanul Kabir Chowdhury.

**Writing – review & editing:** Mohiuddin Ahsanul Kabir Chowdhury, Farhana Karim, Mohammad Mehedi Hasan, Nazia Binte Ali, Abdullah Nurus Salam Khan, Md. Shahjahan Siraj, S. M. Monirul Ahasan, Dewan Md. Emdadul Hoque.

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
