## [Decision Letter · Decision Letter 0]

7 Jan 2022

PONE-D-21-18015Bottleneck Analysis of Maternal and Newborn Health Services in Hard-to-reach areas of Bangladesh using ‘TANAHASHI’ Framework’: an explanatory mixed method studyPLOS ONE

Dear Dr. Mohiuddin Ahsanul Kabir Chowdhury, 

Thank you for submitting your manuscript to PLOS ONE. After careful consideration, we feel that it has merit but does not fully meet PLOS ONE’s publication criteria as it currently stands. Therefore, we invite you to submit a revised version of the manuscript that addresses all the points raised by the two reviewers during the review process.

Please submit your revised manuscript by 21 February 2022 or earlier.  If you will need more time than this to complete your revisions, please reply to this message or contact the journal office at plosone@plos.org. Please include the following items when submitting your revised manuscript:A rebuttal letter that responds to each point raised by the academic editor and reviewer(s). You should upload this letter as a separate file labeled 'Response to Reviewers'.A marked-up copy of your manuscript that highlights changes made to the original version. You should upload this as a separate file labeled 'Revised Manuscript with Track Changes'.An unmarked version of your revised paper without tracked changes. You should upload this as a separate file labelled 'Manuscript'.

We look forward to receiving your revised manuscript.

Kind regards,

Gouranga Lal Dasvarma, PhD

Academic Editor

PLOS ONE

Journal Requirements:

Furthermore, when reporting the results of qualitative research, we suggest consulting the COREQ guidelines: http://intqhc.oxfordjournals.org/content/19/6/349. In this case, please consider including more information on the number of interviewers, their training and characteristics; and please provide the interview guide used.

Additional Editor Comments (if provided):

Please address the comments and suggestions of the two reviewers satisfactorily.

Reviewers' comments:

Reviewer's Responses to Questions

**Comments to the Author**

1. Is the manuscript technically sound, and do the data support the conclusions?

Reviewer #1: Yes

Reviewer #2: Partly

2. Has the statistical analysis been performed appropriately and rigorously? 

Reviewer #1: Yes

Reviewer #2: No

3. Have the authors made all data underlying the findings in their manuscript fully available?

Reviewer #1: Yes

Reviewer #2: No

4. Is the manuscript presented in an intelligible fashion and written in standard English?

Reviewer #1: Yes

Reviewer #2: No

5. Review Comments to the Author

Reviewer #1: Chowdhury and co-authors conducted a Bottleneck Analysis of Maternal and Newborn Health (MNH) Services in Hard-to-reach (HtR) areas of Bangladesh using ‘TANAHASHI’ Framework’ in order to explore the possible remedial approaches. They explored five domains of the framework: availability, accessibility, utilization, adequate coverage and effective coverage by using an explanatory mixed method research approach and collected data from RDWs and stockholders. The study design is appropriate to address the stated research objectives. The data collection tools were the qualitative interviews (55) and structured questionnaire(n=2,768). The data were analyzed using descriptive statistics and thematic approach. Qualitative interviews of facility managers, local stakeholders, RDWs and health care providers (HCP) from village level, union level and sub-district level health facilities were conducted. The study was conducted in four different types of HtR areas (hilly, coastal, lowlands, and river islands). The authors found Antenatal care, under-5 care, and family planning service are available in all facilities while several bottleneck areas namely, availability of drugs; human resources; transportation; lack of knowledge regarding different essential services and health components, and out of pocket expenditure which affected the quality of health care provisions in HtR areas. The manuscript is well written and useful findings are revealed. However, I have a few concerns on the sections, introduction, methods and materials, data analysis and discussion which are given below:

1. Introduction : Authors could describe briefly how the health sytem is functioning in Bangladesh as a background to the strudy that can help reader to know the context and the significance of current research.

Line 56-58: It is good that authors can present the current MMR figure for Bangladesh (i.e.173 per 100000 live births in 2017) along with percent change in MMR during 1990-2016. It is also mentioned that the NMR was reduced from 52 to 30 deaths per 1000 live births between 1994-2017. According to WHO (2018) NMR is 17.1 deaths per 1000 live births. Figures need be checked.

2. Line 73-74: Though the authors assumed that failure in health system in HtR areas, it is not clear how the health situation was measured/ justifications are not clear (i.e., high MMR, NMR in HtR etc.,)

3. Study participants: It was mentioned that the study participants consisted of 2,768 recently delivered women (RDW), however, reference period should be mentioned as one year.

Sampling technique: For the quantitative data collection, authors can elaborate the sampling procedure to reflect sampling frame, sampling method (i.e. random, multi-stage process) that involved when drawing random samples from HtR areas.

5. For the qualitative component, the snowball sampling method was used, authors may explain why this method was employed to recruit participants.

6.Line 166, Data analysis: This could be stated as 'Methods of data analysis'

7. Results: Authors have presented findings of both quantitative and qualitative data pertaining to the five domains:availability, accessibility, utilization, adequate coverage and effective coverage which explained bottelneck areas in facilities in HtR areas. Overall findings address the research gaps and contribute to the exsisting quentum of knowledge.

8. Tables: Table 1, both numbers and percentages (%) are not necessary to present in the same table. if authors use only %s by mentioning number ( n=? in columns ) redear can better grasp the table.

9. It is good that authors could summarize the study findings in a way that can facilitate readers to grasp key findings such as indicating levels of facilities available, its magnitude by comparing national level facilities along with HtR areas.

10. Minor language errors were noted so that it is good that the manuscript could be copy edited before submitting the revised version.

11. Authors have explored a substantive literature related to the area of research and cited them properly.

Reviewer #2: Abstract: Authors might want to write it in structured format.

The authors wrote “Bangladesh being a low-and-middle income country, set targets for MNH aligning with Sustainable 24 Development Goals” ---how does a country’s income level matters here? Rewrite this sentence, I think it does not quite make sense to bring that ‘low-and-middle income’ category as a reason to set SDG targets.

Introduction

• “342 to 211 per 100,000 live births between 2000-2017, still over 800 women are dying from pregnancy and childbirth related complications”—where are these women dying and how frequently? Please specify a country, region or worldwide—where? And annually or over a certain period of time?

• Can authors clearly demonstrate the gap in a table/box? They might want to bring the annual rate of decline in MMR and NMR over the past (e.g. 1990 to 2017). Also include in the table the numbers on where we reach provided the similar rates of decline by 2030, and the rates required to reach the SDG targets.

• Can authors also talk about the gaps in health service coverage nationally vs in HtR areas, both for key maternal and neonatal interventions?

• The detailed description of Tanahashi framework might be omitted from introduction. Authors might want to demonstrate the figure in method section.

Materials and methods

• Did the authors go for 100% of the villages in the 4 HtR district? It is not clear.

• Use a reference for ‘explanatory sequential mixed-method design’.

• How did the data saturation work for different categories of participants? Having 1-2 sentences about authors’ practical insight into it could be useful.

• How was the ‘availability’ determined by interviews or observation of the records from all health facilities of the 4 districts? Please confirmed it. Also, how did you explore clients’ need? How did you define the actual volume of the clients in the study location?

• Please mention the total lists of unions in each HtR district? What is your reason to choose 3 unions from each HtR district?

• How did you define RDW? Is 2989 the all RDWs in the 36 villages?

• Did you select all facility managers and providers in the study areas? If not how did you select them for the interviews? You used snowball for local stakeholders, but how did you get the other qualitative participants in your study?

• In the tools section, you said you had IDIs with RDWs? And FGDs with community people? Can you mention how many IDIs and FGDs conducted in the method above?

• I would suggest you make a simple summary table with type of data collection methods, type of study participants, type of data collection tools, number of participants to make it easier for the readers to understand the types of data and different participants in your study.

• Your overall sampling process for quanti looks like ‘multi-stage convenient sampling’.

• In variable section, your key variables are those stated in Tanahashi framework. Please provide your working definition (clearly mention the operational definition used in your analysis) availability, accessibility, utilisation, and others. I would suggest a simple table again to list each sub variables by those key domain variables (availability, accessibility, utilisation, Adequate coverage and Effective coverage).

• Submit as additional files of all of your data collection tools.

Analysis

• Mention the key descriptive stats utilised.

• Mention the number of transcripts and people involved in analysing qualitative data.

• How did you check inter coder reliability as stated?

• Upload the primary code book and the master code list file as additional files.

• Provide a reason (maybe on method section above) on why FGDs, IDIs, KIIs—all of these needed in your study? There seems a big bulk of data—was it all necessary? If so why, for which aspects of your study? How did your principle of saturation worked out while you were collecting data using these many different methods?

Ethics

• Were the written consents taken from RDWs? There are nearly 15% who had never been to school as per your data. How did it work?

• Did you exclude any categories of RDWs for any ethical reasons? Such as those who had lost their babies recently??

• Did you collect written consents for all qualitative participants too? How did it work for FGDs?

Results

• Can you give a brief description on which level of the 29 health facilities are in your health system? You said only 13.7% were providing CS---were all 29 facilities eligible to provide CS as per your Ministry/Department of Health? The same for blood transfusion ! How many of the 29 facilities are Comprehensive Emergency Obstetric and Newborn Care providers?

• For each component analysis—how does a reader understand how many were supposed to provide that particular care component, say ‘Antenatal care’? For the HR it is clear as you mention the sanctioned posts.

• I would strongly suggest reorganizing the results as per the Tanahashi domain variables—be specific to mention your results as per the key domain variables of the framework you utilised. You can mention the socio-demographic characteristics of study participants—of both quantitative and qualitative participants at the start of the result section. In so doing, you can bring in the key qualitative information integrating together with the key domains of the Tanahashi model.

• Provide a list of key bottlenecks as per each domain in the model, maybe do so in a table. You have collected a lot of qualitative data—I think there could be plenty of info on bottlenecks.

Discussion

Needs further work. Once you have reorganized the result section well as per the model you used—you then can choose key points to interpret, elaborate and provide compelling discussion. Also, acknowledge the limitation of the convenient sampling you have utilised, and strengths of the range of data collection methods employed.

Conclusion: Authors need to write a brief conclusion separately.

Thank you for the great work.

6. PLOS authors have the option to publish the peer review history of their article (what does this mean?). If published, this will include your full peer review and any attached files.

Reviewer #1: **Yes: **Sunethra Perera

Reviewer #2: No

---

## [Author Response · Author response to Decision Letter 0]

3 Mar 2022

RESPONSE TO THE COMMENTS OF THE REVIEWER FOR MANUSCRIPT TITLED “BOTTLENECK ANALYSIS OF MATERNAL AND NEWBORN HEALTH SERVICES IN HARD-TO-REACH AREAS OF BANGLADESH USING ‘TANAHASHI’ FRAMEWORK’: AN EXPLANATORY MIXED METHOD STUDY” [PONE-D-21-18015]

Reviewer #1: Chowdhury and co-authors conducted a Bottleneck Analysis of Maternal and Newborn Health (MNH) Services in Hard-to-reach (HtR) areas of Bangladesh using ‘TANAHASHI’ Framework’ in order to explore the possible remedial approaches. They explored five domains of the framework: availability, accessibility, utilization, adequate coverage and effective coverage by using an explanatory mixed method research approach and collected data from RDWs and stockholders. The study design is appropriate to address the stated research objectives. The data collection tools were the qualitative interviews (55) and structured questionnaire(n=2,768). The data were analyzed using descriptive statistics and thematic approach. Qualitative interviews of facility managers, local stakeholders, RDWs and health care providers (HCP) from village level, union level and sub-district level health facilities were conducted. The study was conducted in four different types of HtR areas (hilly, coastal, lowlands, and river islands). The authors found Antenatal care, under-5 care, and family planning service are available in all facilities while several bottleneck areas namely, availability of drugs; human resources; transportation; lack of knowledge regarding different essential services and health components, and out of pocket expenditure which affected the quality of health care provisions in HtR areas. The manuscript is well written and useful findings are revealed. However, I have a few concerns on the sections, introduction, methods and materials, data analysis and discussion which are given below:

1. Introduction: Authors could describe briefly how the health system is functioning in Bangladesh as a background to the study that can help reader to know the context and the significance of current research.

Line 56-58: It is good that authors can present the current MMR figure for Bangladesh (i.e. 173 per 100000 live births in 2017) along with percent change in MMR during 1990-2016. It is also mentioned that the NMR was reduced from 52 to 30 deaths per 1000 live births between 1994-2017. According to WHO (2018) NMR is 17.1 deaths per 1000 live births. Figures need be checked.

Response: Thank you very much for the comment. We have updated the numbers according to your suggestion [Line 55-60]

2. Line 73-74: Though the authors assumed that failure in health system in HtR areas, it is not clear how the health situation was measured/ justifications are not clear (i.e., high MMR, NMR in HtR etc.,)

Response: Some information has been added related to the issue by the reviewer. [Line 71-75] 

3. Study participants: It was mentioned that the study participants consisted of 2,768 recently delivered women (RDW), however, reference period should be mentioned as one year.

Response: The reference period is added in line 105-106. 

4. Sampling technique: For the quantitative data collection, authors can elaborate the sampling procedure to reflect sampling frame, sampling method (i.e. random, multi-stage process) that involved when drawing Response random samples from HtR areas.

Response: Sampling technique has been specified in the respective section as per the suggestion of the reviewer [Lines 115 and 122]

5. For the qualitative component, the snowball sampling method was used, authors may explain why this method was employed to recruit participants.

Response: The qualitative sampling technique is explained elaborately in lines 125-130

6.Line 166, Data analysis: This could be stated as 'Methods of data analysis'

Response: Thank you for the comment. We have revisited the title as “Data analysis methods” [Line 189]

7. Results: Authors have presented findings of both quantitative and qualitative data pertaining to the five domains: availability, accessibility, utilization, adequate coverage and effective coverage which explained bottleneck areas in facilities in HtR areas. Overall findings address the research gaps and contribute to the existing quantum of knowledge.

Response: Thanks for your kind acknowledgement

8. Tables: Table 1, both numbers and percentages (%) are not necessary to present in the same table. if authors use only %s by mentioning number (n=? in columns) reader can better grasp the table.

Response: Table 1 [Now it is Table 3] has been updated as per your kind suggestion [Line 235-236]

9. It is good that authors could summarize the study findings in a way that can facilitate readers to grasp key findings such as indicating levels of facilities available, its magnitude by comparing national level facilities along with HtR areas.

Response: Thanks for your kind acknowledgement

10. Minor language errors were noted so that it is good that the manuscript could be copy edited before submitting the revised version.

Response: Thanks for your kind concern. The manuscript has been copy-edited.

11. Authors have explored a substantive literature related to the area of research and cited them properly.

Response: Thanks for your kind acknowledgement

Reviewer #2: 

Abstract: Authors might want to write it in structured format.

The authors wrote “Bangladesh being a low-and-middle income country, set targets for MNH aligning with Sustainable 24 Development Goals” ---how does a country’s income level matters here? Rewrite this sentence, I think it does not quite make sense to bring that ‘low-and-middle income’ category as a reason to set SDG targets.

Response: This confusing sentence has been removed.

Introduction

• “342 to 211 per 100,000 live births between 2000-2017, still over 800 women are dying from pregnancy and childbirth related complications”—where are these women dying and how frequently? Please specify a country, region or worldwide—where? And annually or over a certain period of time?

Response: Sentence has been updated [Line 44]

• Can authors clearly demonstrate the gap in a table/box? They might want to bring the annual rate of decline in MMR and NMR over the past (e.g. 1990 to 2017). Also include in the table the numbers on where we reach provided the similar rates of decline by 2030, and the rates required to reach the SDG targets.

Response: We have not added a box/table. However, we have added the suggested information in narratives in the 2nd paragraph of the introduction section [Line 53-65]

• Can authors also talk about the gaps in health service coverage nationally vs in HtR areas, both for key maternal and neonatal interventions?

Response: We have added references that showed the differences between national surveys and findings from the area studied. [Line 70-75]

• The detailed description of Tanahashi framework might be omitted from introduction. Authors might want to demonstrate the figure in method section.

Response: We have added the description of Tanahashi Framework as Supplementary Document 1 according to your kind suggestion. 

Materials and methods

• Did the authors go for 100% of the villages in the 4 HtR district? It is not clear.

Response: We did not go for 100% villages as mentioned in the sampling section where we mentioned that we visited randomly selected 36 villages of these districts. [Line 120-121]

• Use a reference for ‘explanatory sequential mixed-method design’.

Response: Reference added in line 87.

• How did the data saturation work for different categories of participants? Having 1-2 sentences about authors’ practical insight into it could be useful.

Response: An explanation with reference is added in line 108-111.

• How was the ‘availability’ determined by interviews or observation of the records from all health facilities of the 4 districts? Please confirmed it. Also, how did you explore clients’ need? How did you define the actual volume of the clients in the study location?

Response: Availability was measured from supply side perspective, for example according to the available health resources as per the government created positions, availability of services and other resources. However, the indicator matrix [Supplementary Document 4] helped us to determine the indicators for different domains

• Please mention the total lists of unions in each HtR district? What is your reason to choose 3 unions from each HtR district?

Response: Supplementary Document 2 is added for the list of the unions. We chose 3 union to match the budget and the sample size. 

• How did you define RDW? Is 2989 the all RDWs in the 36 villages?

Response: Recently delivered women here were those, whose delivery occurred within one year of data collection period. [Line 105-106]

No, 2989 were not all RDWs in 36 villages, our data collectors reached the centre of each village and then randomly chose a direction by pen-spinning method. Then, the data collectors visited every alternate household to collect information regarding RDWs. This description is added in Sampling technique section. [Line 121-125]

• Did you select all facility managers and providers in the study areas? If not, how did you select them for the interviews? You used snowball for local stakeholders, but how did you get the other qualitative participants in your study?

Response: Thank you very much for your keen observation. We selected the health facility managers and health care providers through purposive sampling. This information is added in line 129-130.

• In the tools section, you said you had IDIs with RDWs? And FGDs with community people? Can you mention how many IDIs and FGDs conducted in the method above?

Response: This information is added in line 202.

• I would suggest you make a simple summary table with type of data collection methods, type of study participants, type of data collection tools, number of participants to make it easier for the readers to understand the types of data and different participants in your study.

Response: Thanks for the suggestion. A table has been added at line 113, Page 6 accordingly

• Your overall sampling process for quanti looks like ‘multi-stage convenient sampling’.

Response: Yes, we have mentioned it in line 115 in revised version. 

• In variable section, your key variables are those stated in Tanahashi framework. Please provide your working definition (clearly mention the operational definition used in your analysis) availability, accessibility, utilisation, and others. I would suggest a simple table again to list each sub variables by those key domain variables (availability, accessibility, utilisation, Adequate coverage and Effective coverage).

Response: All variables have been defined at Supplementary Document 4. 

• Submit as additional files of all of your data collection tools.

Response: The Data collection tools have been added as Supplementary Document 3.

Analysis

• Mention the key descriptive stats utilized.

Response: It is now mentioned in Line 192-194

• Mention the number of transcripts and people involved in analysing qualitative data.

Response: The numbers are mentioned in table 3 [Line 113], and qualitative analysis section is revisited [Line 179-184]. We have also added a sub-section “Data collectors’ training and data collection” (167-188)

• How did you check inter coder reliability as stated?

Response: Thanks for the comment. We have added an explanation regarding how we check inter-coder reliability [Line 209-212]

• Upload the primary code book and the master code list file as additional files.

Response: We have Shared our codebook as supplementary Document 5.

• Provide a reason (maybe on method section above) on why FGDs, IDIs, KIIs—all of these needed in your study? There seems a big bulk of data—was it all necessary? If so why, for which aspects of your study? How did your principle of saturation worked out while you were collecting data using these many different methods?

Response: We have added the explanation in the method section [Line 139-147]

Ethics

• Were the written consents taken from RDWs? There are nearly 15% who had never been to school as per your data. How did it work?

Response: Regarding participants who had no formal education, their thumb impression was collected as consent. In line 217-221 of revised version, this information has been added. 

• Did you exclude any categories of RDWs for any ethical reasons? Such as those who had lost their babies recently??

Response: No, we did not exclude any categories of RDWs for any ethical reasons. However, we did not include those RDW who had stillbirth in this study data collection because we were also collecting information about newborn care.

• Did you collect written consents for all qualitative participants too? How did it work for FGDs?

Response: We have added this information under Ethical consideration. [Line 217-221]

Results

• Can you give a brief description on which level of the 29 health facilities are in your health system? You said only 13.7% were providing CS---were all 29 facilities eligible to provide CS as per your Ministry/Department of Health? The same for blood transfusion! How many of the 29 facilities are Comprehensive Emergency Obstetric and Newborn Care providers?

Response: Thank you very much for your knowledgeable observation. According to Ministry/Department of Health of Bangladesh, only 10 UHCs are CEmONC facilities in the study, hence C-section service and blood transfusion services supposed to be present in 10 UHCs, and the table has been revisited. [Line 236, table 3]

• For each component analysis—how does a reader understand how many were supposed to provide that particular care component, say ‘Antenatal care’? For the HR it is clear as you mention the sanctioned posts.

Response: We analyzed as per our indicator matrix presented in supplementary table 4, and the denominator of the calculation for availability of human resources was the sanctioned posts for availability of HRs

• I would strongly suggest reorganizing the results as per the Tanahashi domain variables—be specific to mention your results as per the key domain variables of the framework you utilised. You can mention the socio-demographic characteristics of study participants—of both quantitative and qualitative participants at the start of the result section. In so doing, you can bring in the key qualitative information integrating together with the key domains of the Tanahashi model.

Response: Result section has been reorganized according to your kind suggestion

• Provide a list of key bottlenecks as per each domain in the model, maybe do so in a table. You have collected a lot of qualitative data—I think there could be plenty of info on bottlenecks.

Response: Table 6 has been added at Line 318

Discussion

Needs further work. Once you have reorganized the result section well as per the model you used—you then can choose key points to interpret, elaborate and provide compelling discussion. Also, acknowledge the limitation of the convenient sampling you have utilised, and strengths of the range of data collection methods employed.

Response: The discussion is revised as per your advice and several paragraphs have been added. We have also added a sub-section as “Strengths and Limitations” where your feedback was addressed.

Conclusion: Authors need to write a brief conclusion separately.

Response: A conclusion is added

Thank you for the great work.

Response: Thank you very much for your kind review as well. It really helped to improve the manuscript.

---

## [Editor Report · Decision Letter 1]

30 Mar 2022

PONE-D-21-18015R1Bottleneck Analysis of Maternal and Newborn Health Services in Hard-to-reach areas of Bangladesh using ‘TANAHASHI’ Framework’: an explanatory mixed method studyPLOS ONE

Dear Dr. Mohiuddin Ahsanul Kabir Chowdhury,

Thank you for submitting your manuscript to PLOS ONE. After careful consideration, we feel that it has merit but does not fully meet PLOS ONE’s publication criteria as it currently stands. Therefore, we invite you to submit a revised version of the manuscript that addresses the points raised during the review process.

We look forward to receiving your revised manuscript.

Kind regards,

Gouranga Lal Dasvarma, PhD

Academic Editor

PLOS ONE

Journal Requirements:

Additional Editor Comments:

Dear Dr. Mohiuddin Ahsanul Kabir Chowdhury,

• It appears you have changed your affiliation to Asian University 10 for Women, Chittagong, Bangladesh since submitting the manuscript. The manuscript, if published will bear the name of the institute of your current affiliation, but you should acknowledge (in the acknowledgement section) that the work on the manuscript/paper was carried when you were working at the James P Grant School of Public Health, BRAC University, Dhaka, Bangladesh. The same comment applies to your c-author S M Monirul Ahasan, who has changed his affiliation from International Rescue Committee to World Vision.

• A thorough, professional editing for English is required before the manuscript is accepted for publishing. A few examples are given below, but have the entire manuscript edited.

Line 70 of Manuscript Revision#1: “…mortality rates have been increased…”. Delete the word “been”. Or, do you mean “…mortality rates will have increased…”?

Line 71: Re “neo-normal era”. How is this neo-normal era when Covid-19 is still raging?

Line 75: “…health services by poor population…”. Should be “…by the poor population…”

Reviewer#1:

• Table 3 (previously Table 1) shows the results for 29 facilities, but you selected 36 villages. Did all the selected villages not have a facility?

• Line 73-74: Though the authors assumed that failure in health system in HtR areas, it is not clear how the health situation was measured/ justifications are not clear (i.e., high MMR, NMR in HtR etc.,)

Academic Editor: Your response does not address the reviewer’s comments entirely. Please re-read the comments and address them appropriately.

Reviewer#2:

• “Can authors talk about the gaps in health service coverage nationally vs in HtR areas, both for key maternal and neonatal interventions?”

Academic Editor: Your response only states that gaps exist. But please state the extent of the gaps by stating respective figures.

• “Materials and methods”:

“Did the authors go for 100% of the villages in the 4 HtR district? It is not clear”.

Response: We did not go for 100% villages as mentioned in the sampling section where we mentioned that we visited randomly selected 36 villages of these districts. [Line 120-121]

Academic Editor: Please state why you decided to take 36 villages.
---

## [Author Response · Author response to Decision Letter 1]

15 Apr 2022

RESPONSE TO THE COMMENTS FROM THE REVIEWER AND ACADEMIC EDITOR

• It appears you have changed your affiliation to Asian University for Women, Chittagong, Bangladesh since submitting the manuscript. The manuscript, if published will bear the name of the institute of your current affiliation, but you should acknowledge (in the acknowledgement section) that the work on the manuscript/paper was carried when you were working at the James P Grant School of Public Health, BRAC University, Dhaka, Bangladesh. The same comment applies to your c-author S M Monirul Ahasan, who has changed his affiliation from International Rescue Committee to World Vision.

RESPONSE: 

We have added an acknowledgement section where we mentioned the requested information

• A thorough, professional editing for English is required before the manuscript is accepted for publishing. A few examples are given below, but have the entire manuscript edited.

Line 70 of Manuscript Revision#1: “…mortality rates have been increased…”. Delete the word “been”. Or, do you mean “…mortality rates will have increased…”?

Line 71: Re “neo-normal era”. How is this neo-normal era when Covid-19 is still raging?

Line 75: “…health services by poor population…”. Should be “…by the poor population…”

RESPONSE: 

We have copy-edited the manuscript which can be seen in the track changes version. 

Reviewer#1:

• Table 3 (previously Table 1) shows the results for 29 facilities, but you selected 36 villages. Did all the selected villages not have a facility?

RESPONSE: 

No, not every village had its separate health facility. We have added a sentence regarding this at the end of study site section. (Line 107-108 )

• Line 73-74: Though the authors assumed that failure in health system in HtR areas, it is not clear how the health situation was measured/ justifications are not clear (i.e., high MMR, NMR in HtR etc.,)

Academic Editor: Your response does not address the reviewer’s comments entirely. Please re-read the comments and address them appropriately.

RESPONSE: 

We have added a couple more references and narratives regarding the situation in HtR in Bangladesh. However, the information regarding HtR areas are sparse, and this also adds strength to our study. (Line 75-80)

Reviewer#2:

• “Can authors talk about the gaps in health service coverage nationally vs in HtR areas, both for key maternal and neonatal interventions?”

Academic Editor: Your response only states that gaps exist. But please state the extent of the gaps by stating respective figures.

RESPONSE: 

We have not included the figure since there would be many, rather we have included a sentence regarding this, and added a reference. (Line 77-80)

• “Materials and methods”:

“Did the authors go for 100% of the villages in the 4 HtR district? It is not clear”.

Response: We did not go for 100% villages as mentioned in the sampling section where we mentioned that we visited randomly selected 36 villages of these districts. [Line 120-121]

Academic Editor: Please state why you decided to take 36 villages.

RESPONSE: 

We could not visit all the villages due to budget constraints, and selected 36 to attain our required sample size

---

## [Editor Report · Decision Letter 2]

21 Apr 2022

Bottleneck Analysis of Maternal and Newborn Health Services in Hard-to-reach areas of Bangladesh using ‘TANAHASHI’ Framework’: an explanatory mixed method study

PONE-D-21-18015R2

Dear Dr. Mohiuddin Ahsanul Kabir Chowdhury,

We’re pleased to inform you that your manuscript has been judged scientifically suitable for publication and will be formally accepted for publication once it meets all outstanding technical requirements.

Kind regards,

Gouranga Lal Dasvarma, PhD

Academic Editor

PLOS ONE

Additional Editor Comments (optional):

Thank you for addressing the final set of comments. The manuscript may now be considered for publication.
---

## [Editor Report · Acceptance letter]

27 Apr 2022

PONE-D-21-18015R2 

Bottleneck Analysis of Maternal and Newborn Health Services in Hard-to-reach areas of Bangladesh using ‘TANAHASHI’ Framework’: an explanatory  mixed-method study 

Dear Dr. Chowdhury:

I'm pleased to inform you that your manuscript has been deemed suitable for publication in PLOS ONE. Congratulations! Your manuscript is now with our production department. 

Kind regards, 

on behalf of

Dr. Gouranga Lal Dasvarma 

Academic Editor

PLOS ONE